# Associations of Birth Size with Physical and Cognitive Function in Men and Women 60 Years and Older—Systematic Review and Meta-Analysis

**DOI:** 10.3390/nu17162583

**Published:** 2025-08-08

**Authors:** Vilborg Kolbrun Vilmundardottir, Birna Thorisdottir, Alfons Ramel, Ólöf Guðný Geirsdóttir

**Affiliations:** 1Faculty of Food Science and Nutrition, University of Iceland, 102 Reykjavik, Iceland; vkv2@hi.is (V.K.V.); bith@hi.is (B.T.); alfonsra@hi.is (A.R.); 2Unit for Nutrition Research, Landspitali-University Hospital, University of Iceland, 102 Reykjavik, Iceland

**Keywords:** birth size, life course, healthy aging, cognitive function, physical function

## Abstract

Background/Objectives: This systematic review aimed to investigate the relationship between birth size, a marker of prenatal undernutrition, and both physical and cognitive function in individuals aged 60 years and older. Methods: We searched the PubMed and Scopus databases up to November 2024 for prospective cohort studies that included data on birth size and physical or cognitive function in individuals aged 60 or older, excluding studies focused on preterm individuals. The Risk Of Bias In Non-randomized Studies—of Exposure (ROBINS-E) tool was used to evaluate the bias of each included study. Fixed-effects meta-analysis was performed using Review Manager. This systematic review was registered with PROSPERO, CRD42023360823. Results: Twenty-four articles met the eligibility criteria, with participant numbers ranging from 52 to 4000 (about 50% women) and an average age range of 60.9 to 78.4 years. Eight articles had a high risk of bias, while the remaining 16 presented some concerns. Three meta-analyses were conducted: two for grip strength and one for word fluency. Grip strength was positively associated with birth weight, both in an unadjusted analysis, which showed an increase of 1.88 kg (95% CI 1.19, 2.56), and in an analysis adjusted for age, sex, and height/body mass index, which showed an increase of 1.15 kg (95% CI 0.71, 1.59). Word fluency also displayed a positive association with birth weight, with an increase of 0.62 words per minute (95% CI 0.15, 1.10). Conclusions: Smaller birth size, indicative of prenatal undernutrition, is associated with diminished physical and cognitive function in later life. These results highlight the importance of identifying individuals born small as a vulnerable group and implementing lifelong strategies to promote healthy aging.

## 1. Introduction

Quality nutrition throughout the life course is one of the key factors in maintaining optimal health and quality of life, from gestation to older age [1,2]. In particular, the concept of healthy aging, defined not merely by the absence of disease but by the maintenance and optimization of intrinsic capacity throughout life, has become increasingly important as the global population ages. The World Health Organization (WHO) projects that the proportion of people aged 60 years or older will nearly double from 12% in 2015 to 22% in 2050 [3]. In this context, it is crucial to further understand the early life determinants of intrinsic capacity, which encompasses all the physical and cognitive capacities of an individual and serves as a foundation for healthy aging [4,5].

Among these early life factors, birth size, in particular birth weight, has been considered a possible marker of prenatal environment, with smaller birth size suggesting prenatal undernutrition or an otherwise suboptimal fetal environment, such as mothers’ health, social status, etc. [1,6]. Prenatal undernutrition can have long-term consequences, where lower birth weight has been associated with a greater risk of coronary heart disease and type 2 diabetes [7,8].

Beyond disease risk, birth size has also been associated with measures such as body composition, with individuals born small showing altered fat distribution and reduced muscle mass in infancy until middle age [9,10]. Further, considering physical function, a systematic review and meta-analysis on individuals aged 5 to 68 years found that for each kg increase in birth weight, grip strength increased by 2.07 kg (95%CI 1.47, 2.66) in men and 1.59 kg (95%CI 1.25, 1.93) in women [11].

Similarly, associations have been reported between smaller birth size and poorer cognitive function, including lower cognitive ability, reduced language skills, and slower simple reaction time during adolescence and young adulthood [12,13,14]. Reinforcing these findings, a systematic review and meta-analysis involving participants aged 3 to 22 years found that individuals with low birth weight (<2.5 kg) had significantly lower IQ scores, by an average of 7.63 points (95%CI 5.95, 9.31) compared to those with normal birth weight (≥2.5 kg) [15]. Moreover, a recent systematic review suggested that smaller birth size may be associated with poorer cognitive functioning among individuals aged 50 years and older [16].

Given the projected increases in older populations and the emphasis on optimizing cognitive and physical function for healthy aging, it is imperative to expand knowledge from early/middle age and into older age on how prenatal factors such as birth size can influence cognitive and physical function in later life.

This systematic review aims to investigate the association between birth size and physical and cognitive function in men and women aged 60 years and older, motivated by the need to understand early life determinants of health in an aging population.

## 2. Materials and Methods

This systematic review was conducted following the Preferred Reporting Items for Systematic Reviews and Meta-Analyses (PRISMA) statement. A study protocol was published before article selection in the PROSPERO database. ^©^Centre for Reviews and Dissemination, University of York, 2008 Published by CRD, University of York January 2009 ISBN 978-1-900640-47-3. (https://www.crd.york.ac.uk/PROSPERO/view/CRD42024559392 (accessed on 8 June 2025)).

### 2.1. Eligibility Criteria

Eligible participants were men and women aged 60 years and older who were born full term (≥37 weeks’ gestational age). Studies with exclusively preterm-born participants were excluded. However, a preliminary search by the primary author revealed that restricting the population to exclusively term-born participants yielded very few studies on the associations between the exposures and outcomes of interest. Therefore, studies on population-based samples (including mostly term-born participants) were included. The primary exposure of interest was birth weight, presented as either a continuous or categorical variable (with a comparator set as higher versus lower). Other exposures of interest included birth length, ponderal index, and body mass index at birth. The outcomes of interest were physical function and cognitive function. They could be assessed in various ways, including the short physical performance battery (SPPB), grip strength, timed-up-and-go, gait speed, leg strength, sarcopenia, frailty, memory, executive function, processing speed, Mini-Mental State Examination (MMSE), brain images (MRI, CT), and diagnosed impaired cognitive function. Only prospective cohort studies were deemed eligible for inclusion.

### 2.2. Search Strategy 

A comprehensive literature search was conducted on PubMed and Scopus at the University of Iceland, initially in January 2023 and subsequently updated in November 2024. The primary author and a research librarian at the University of Iceland developed the search strategy (Appendix A). The reference lists of relevant retrieved articles were screened to identify additional relevant articles. There were no publication dates or language limitations in the search. Grey literature and unpublished study searches were not performed. 

### 2.3. Article Selection 

Two authors (VKV and BT) independently screened and selected studies for inclusion. The screening of title/abstracts and full-text articles was performed in the web tool Covidence (https://www.covidence.org/ (accessed on 10 January 2025)). Discrepancies were resolved through discussion among the study team, both after title/abstract screening and full-text screening. 

### 2.4. Data Collection 

Data from full-text articles were extracted in standardized extraction forms by two authors (VKV and BT) working independently. Among the data extracted were information on participants (including sex, age, and sample size) and settings, eligibility and recruitment, exclusion reasons, assessment of exposures, evaluation of outcomes, confounding variables, results (unadjusted and adjusted estimates), study authors’ conclusions, reported conflict of interest by study authors, and funding source. Discrepancies were resolved by discussion.

### 2.5. Risk of Bias Assessment 

The Risk Of Bias In Non-randomized Studies of Exposure (ROBINS-E) tool (https://www.riskofbias.info/welcome/robins-e-tool (accessed on 5 February 2025)) was used to assess the risk of bias (RoB) of each included study as low, some, or high by two authors (BT and AR) working independently. Discrepancies were resolved by discussion of the study team. The ROBINS-E tool considers the risk of bias in seven domains: due to confounding (we used Variant A), arising from measurements of the exposure (we used Variant A), in selection of participants, due to post-exposure interventions, due to missing data, arising from measurement of outcomes, and in the selection of the reported result. Decisions were made prior to the RoB assessment. Age and sex were considered confounding factors that needed to be controlled for, to avoid introducing reporting bias due to confounding (Domain 1). Additionally, physical activity was considered a key confounding factor for physical function outcomes, while education and socio-economic status were considered key confounding factors for cognitive function outcomes. All studies except those describing samples consisting of only term-born participants were considered to introduce some risk that the measured exposure would not well-characterize the exposure specified to be of interest in this systematic review (Domain 2). The Risk of bias was visualized by using the web app Risk-Of-Bias VISualization (robvis) (https://www.riskofbias.info/welcome/robvis-visualization-tool (accessed on 5 February 2025)). 

### 2.6. Synthesis Methods 

The physical function outcomes were classified into eleven groups, based on the function being measured, as described in the methods chapters of included articles: grip strength, performance score, walking, chair rise, balance, arm curl, lower flexibility, upper flexibility, cardiorespiratory fitness (VO_2max_), self-reported function and frailty (Appendix A). The cognitive function outcomes were classified into twelve groups based on the cognitive domain being measured [17,18]: cognitive impairment, executive function, word fluency, cognition, verbal memory, brain volume, processing speed, cognitive decline, simple attention, intelligence, premorbid ability, and visual memory (Appendix A). As some articles reported results for several cognitive outcomes that tested similar/same cognitive domains, some cognitive domains include multiple outcomes from the same article.

Meta-analyses were performed if deemed appropriate to combine/pool the different studies, i.e., if two or more cohort studies with sufficient homogeneous data existed [19]. These conditions were met for two outcomes: grip strength [20,21,22] and verbal fluency [23,24,25]. Confidence intervals were calculated from β- and *p*-values in one study, as they were not reported [23]. A fixed-effects meta-analysis with the generic inverse variance method was conducted using Review Manager (RevMan; The Cochrane Collaboration, 2020), version 5.4.1. Potential heterogeneity between studies was quantified using the I^2^ statistic, which estimates (range 0–100%) the proportion of variance in the pooled estimates attributable to differences in estimates between studies included in the meta-analyses. However, due to the low number of included articles in the Meta-analyses, the I^2^ statistic was interpreted with caution, alongside visual forest plots and 95% confidence intervals [26]. Pooled associations were shown using forest plots. Due to the low number of included articles, the risk of publication bias, as assessed by funnel plots, could not be determined.

## 3. Results

As shown in Figure 1, the database searches yielded 768 records after deduplication, of which 734 were excluded following title/abstract screening. The remaining 34, plus eight additional records identified by screening reference lists of retrieved articles and other systematic reviews, were screened in full, after which 18 additional reports were excluded (see articles excluded after full-text screening in Appendix A). Therefore, 24 articles met the inclusion criteria for the review, and six were included in the quantitative synthesis (meta-analysis). The included articles report results from 12 unique prospective cohorts from ten countries, of which all but two (China and India) are classified as high-income countries (Table 1).

### 3.1. Study Characteristics

In total, 10 articles reporting possible observational associations between birth size and physical function and 14 articles reporting possible observational associations between birth size and cognitive function were included (Table 2). The participants were obtained from generally healthy populations, and participant numbers ranged from 52 to 4000, with around 50% women and 50% men. The overall mean age range for all studies was 60.9 to 78.4 years. Most studies (*n* = 21) were publicly funded, while the remaining three did not disclose their funding sources.

Birth weight was available as an exposure in all articles except for two [23,28]. Birth size was obtained from birth records in all articles except two, where birth weight was self-reported [23,44]. Most studies adjusted for age, sex, and cognitive function. Eight studies made further adjustments for education and/or socio-economic status [23,24,27,30,36,40,41,42]. Four studies on physical function adjusted for physical activity [20,23,29,38].

### 3.2. Risk of Bias in Included Studies

RoB assessment is shown in Figure 2, and study-level RoB assessments are shown in Figure 3.

A total of 16 studies were evaluated, of which eight exhibited a high risk of bias (RoB). Most of these studies were assessed as having either some degree of risk or a high risk of bias primarily due to limitations in the measurement of exposure. This aspect is addressed in Domain 2, which examines the accuracy and reliability of how the intervention or exposure was quantified or classified. Additionally, selection bias, analyzed in Domain 3, pertains to the methods employed for participant selection and the potential for these processes to introduce bias into the findings. Notably, the two studies categorized as having a low risk of selection bias were based on registry data in the context of cohort studies.

### 3.3. Summary of Findings

#### 3.3.1. Physical Function

Findings for the eleven different outcome groups are summarized in Table 3 (the outcome groups are further explained in Appendix A). Kuh et al. reported a 1.96 kg (95%CI 1.08, 2.84) increase in GS for men and a 1.07 kg (95%CI 0.25, 1.88) increase in GS for women per kg BW, after adjusting for age and height [21]. Ylihärsila et al. reported a 1.30 kg (95%CI 0.10, 2.50) increase in GS per 1 kg increase in BW, after adjusting for age and BMI [22]. Sayer et al. found a significant correlation between BW and GS (*p* = 0.01) after adjusting for age, sex, height, and social class now and at birth [37]. Sayer et al. found a significant correlation between BW and GS of 0.19 (*p* < 0.001) in men and of 0.16 (*p* < 0.001) in women, after adjusting for age, PA, social class, smoking, and alcohol [38].

Eriksson et al. found a positive association between z-scores of BW 0.067 (95%CI 0.012, 0.123), BL 0.063 (95%CI 0.006, 0.121), and BMI 0.057 (95%CI 0.002, 0.112) and performance Z-score, after adjusting for several confounders (listed in Table 2) [32]. The same authors found the arm curl test to be associated with BL, with a 0.078 (95%CI 0.017, 0140) increase in Z-score per Z-score BL, as well as lower flexibility being associated with BW, with a 0.078 increase in Z-score per z-score BW, after adjusting for previously mentioned confounders and adding adult anthropometry. Martin et al. found a negative association of BW with chair rises in women, with a 0.17 (95%CI 0.05, 0.29) SDS increase in chair rise time per SDS increase in BW, after adjusting for age, height, and weight adjusted for height [39]. The same authors also found a positive association of BW with balance in men, with an odds ratio of 0.68 (95%CI 0.50, 0.91) of being in the lowest balance group (losing balance in one-legged test in less than 5 sec) per SDS BW, after adjusting for previously mentioned confounders [39]. von Bonsdorff et al. found a positive association between BW and self-reported function, where those born < 2.5 kg had a 2.73 OR (95%CI 1.57, 4.72) and those born 2.5 to 2.9 kg had a 1.5 OR (95%CI 2.20, 2.04) of lower functioning when compared to those born 3.0 to 3.4 kg, after adjusting for several confounders (listed in Table 2) [28]. Haapanen et al. found a positive association between BW, BL, and BMI and frailty, where a 1 kg increase in BW was associated with a relative risk ratio of 0.4 for frailty when compared to non-frailty [33]. No associations were found between BW, BL, or BMI and walking, upper flexibility, and VO_2max_ [29,32,39].

The pooled associations between BW and GS were calculated using unadjusted models (Figure 4) and models adjusted for age, height, or BMI (Figure 5). The meta-analysis on unadjusted statistical models used three data sets from two articles (participants, *n* = 2153); Bleker [20] and Yliharsila (reporting results for men and women separately) [22]. The meta-analysis on adjusted statistical models used five data sets from three articles (participants, *n* = 4251), Kuh (reporting results for men and women separately), Bleker, and Yliharsila (reporting results for men and women separately) [20,21,22]. Both approaches indicated that higher BW was significantly associated with higher GS, although statistical adjustments weakened this association. With unadjusted pooled associations showing an increase of 1.88 (95%CI 1.19, 2.56) kg GS per kg BW and adjusted showing an increase of 1.15 (95%CI 0.71,1.59) kg GS per kg BW. The heterogeneity between studies was low to moderate (Figure 4 and Figure 5).

#### 3.3.2. Cognitive Function

Findings for the twelve different outcome groups are summarized in Table 4 (the outcome groups are further explained in Appendix A). Mosing et al. found an increased risk of 1.73 (95% CI 1.00, 1.99) for cognitive impairment in those born small for gestational age, after adjusting for several confounders (listed in Table 2) [41].

Krishna et al. found a positive association between BW and word fluency, with an increase of 0.95 (95%CI 0.24, 1.60) words mentioned/minute per 1 kg increase in BW, after adjusting for age, sex, and sibship [24]. The same authors found a positive association between BW 0.29 (95%CI 0.12, 0.46), BL 0.03 (95%CI 0.00, 0.05), and cognition score, after adjusting for previously mentioned confounders. A positive association between BW and two measures of verbal memory, immediate recall with a 0.82 (95%CI 0.14, 1.50) increase in score per SD BW, and delayed recall with a 0.37 (95%CI 0.58, 1.15) increase in score per SD BW, after adjustments. Lastly, BL was positively associated with delayed recall, with a 0.06 (95%CI 0.01, 0.11) increase in score per SD of BL, after adjustments.

Paile-Hyvärinen et al. found a negative linear association between BW and two measures of processing speed, with a −3.8% (95%CI −6.5, −1.1) in reaction time and −1.5% (95%CI −0.1, −2.9) in hit rate per kg increase in BW, after adjusting for age, sex, and education [27]. Raikkonen et al. found an association in men between BL and cognitive decline, with a 0.10 (95%CI 0.02, 0.18) SD decrease in cognitive ability per SD BL, after adjusting for several confounders (as listed in Table 2) [30]. As well as an association of BW 1.31 (95%CI 0.06, 2.55) and BL 1.43 (95%CI 0.27, 2.58) with intelligence (standardized cognitive ability).

De Rooij et al. found a positive association between BW and intracranial volume (ICV) and total brain volume (TBV) (*p* < 0.05) in an unadjusted model. Muller et al. found a positive association between a decrease in ponderal index and ICV −7.5 mL (−13.7 to −7.0), TBV −3.3 mL (95%CI −6.5, −0.1), white matter volume −1.8 mL (95%CI −3.5, −0.1) and gray matter volume −1.7 mL (95%CI −4.3 to 0.8), after adjusting for several confounders (listed in Table 2) [42]. As well as between BW −20.3 mL (−26.5, −14.1) and BL −13.3 (−19.5, −7.0) and ICV. Walhovd et al. found a positive association between birth weight and cortical volume, where a 1 SD decrease in BW corresponded to an 8466 mm^3^ reduction in cortical volume [44].

Erickson et al. found a positive association in women between BW and a measure of simple attention (Serial 7′s), with a 0.08 (*p* = 0.04) worse scoring per pound of BW. No significant associations were found for executive function, premorbid ability, and visual memory [23,25,40,42].

The pooled associations between BW and verbal fluency were calculated based on data from three articles (Krishna, Erickson, Skogen), with a total of 1359 participants [23,24,25]. All three studies indicated some increased verbal fluency associated with increased BW, and the pooled estimate of 0.62 words/minute (95%CI 0.15, 1.10) was significant, although some heterogeneity between studies was observed (Figure 6).

## 4. Discussion

The current findings suggest that small birth size as an indicator of prenatal undernutrition is associated with both physical and cognitive functioning in older age. This systematic review, to our knowledge, is the first to examine two key components of intrinsic capacity and review the evidence from a life course perspective on healthy aging, thereby expanding findings from birth into older age.

Previous studies have shown associations of small birth size with various cognitive and physical functions in early life and into adulthood [1,9,10,13]. Results of articles included in the current systematic review point to birth size possibly being associated with different cognitive domains, including language skills (such as word fluency), processing speed, and intelligence [24,27,30]. As for physical function, included articles found birth size to be associated with strength, flexibility, balance, self-reported physical function, and frailty [22,28,32,33,37,38]. On the contrary, Martin et al. found a significant negative association of birth weight with chair rises in women. However, the authors suggest it could be a chance finding [39].

Our findings align with those of a prior systematic review and meta-analysis examining the association between birth weight and grip strength measured between 5 and 68 years of age, which showed an increase in grip strength with higher birth weight [11]. Associations between smaller birth size and reduced muscle strength may be rooted in muscle formation during gestation, i.e., exposure to prenatal undernutrition may lead to both lower muscle mass and lower muscle quality later in life [45]. This results in lower muscle strength and altered force generation during muscle contraction. Grip strength is a highly validated method of measuring physical performance, and low grip strength is associated with a poorer overall outlook, including an increased risk of mortality, frailty, sarcopenia, and cognitive decline [46].

Similar to the current findings on cognitive function, a recent systematic review looking at birth size and cognitive ability from 50 years of age, which included many of the same articles as reviewed here (*n* = 8), concluded that smaller birth size can possibly predispose individuals to some cognitive problems [16]. Furthermore, our findings on word fluency are consistent with those of previous studies in younger populations, ranging from adolescence to middle age. Costa et al. found a linear trend between birth weight and word fluency (*p* = 0.004) [12,47]. However, the current results should be interpreted with caution, as some heterogeneity was detected, and the three articles included in the analysis were estimated to have a high risk of bias.

One key to healthy aging may lie in quality nutrition during gestation and throughout the life course, where meeting needs for macro- and micronutrients by eating a balanced diet is a well-established key factor in optimizing health and well-being [1,2]. In the context of birth size, one systematic review and meta-analysis on maternal dietary patterns and birth outcomes found dietary patterns rich in refined grains, processed meat, saturated fat and sugar to be associated with lower birth weight (−40 g 95% −61, −20), whilst finding a weak (unsignificant) protective trend for healthy dietary patterns and risk of low birth weight [48]. Also, studies from the Dutch Famine cohort have found that exposure to famine during gestation increases risk for several worse health outcomes, independent of birth size [49]. While monitoring maternal and newborn weight is a straightforward and cost-effective method to assess nutritional status, it may be beneficial to shift the primary focus from weight measurements to evaluating nutritional intake. Established validated tools are available for this purpose, offering insights into potential deficiencies during gestation. This approach allows for timely interventions and helps ensure food security [50]. Ultimately, providing such measures by observing nutritional status and ensuring quality nutrition throughout the life course could be a stepping stone towards healthy aging.

### Strengths and Limitations

A strength of this systematic review is the established process for undertaking robust systematic reviews. A protocol was registered prior to the review process to enhance transparency in the review process. We searched two relevant databases to identify articles appropriate to the topic. These databases are considered to cover most of the literature on the topic of interest. In addition to a comprehensive database search, we screened the reference lists of included articles and other systematic reviews to identify possible additional articles of interest. Therefore, we consider it unlikely that we missed any relevant literature. Furthermore, the review processes were rigorously implemented, with independent assessments by two authors taken at every stage, including literature screening, data extraction, and risk of bias assessment.

Using birth size as a measure of prenatal undernutrition is a limitation due to its indirect nature and influence by multiple factors. As the exposure of interest, prenatal undernutrition, is difficult to measure; therefore, we used birth size as a proxy for it. However, prenatal undernutrition can have varying effects on birth size, depending on several factors, including the timing and duration of undernutrition. Additionally, several factors beyond prenatal undernutrition influence birth size. We attempted to minimize the effect of gestational age at birth, on birth size by excluding studies on preterm-born participants and considering gestational age for other cohorts in the risk of bias assessment. While we acknowledge that the exposure of interest has its limitations, we believe it to be the best possible choice.

In examining the outcomes of interest, it is notable that the metrics employed in studies addressing cognitive outcomes exhibited substantial heterogeneity. This variability complicates direct comparisons between articles and ultimately culminates in a situation where only a single outcome, namely verbal fluency, qualifies for inclusion in a meta-analysis. To address this and facilitate the reporting of results, we grouped them roughly by cognitive domains being measured by each test, as described by the authors or available information on neuropsychological testing [17,18]. As for physical function outcomes, they were sometimes measuring the same function, but due to different tests being used, they were too heterogeneous for pooling associations. For example, Eriksson et al. used a walking test where participants walked for 6 min and the distance was measured, whilst Martin et al. used a test where participants walked three meters for time [32,39], thus only allowing for single article results to be reported.

## 5. Conclusions

Our findings provide evidence that small birth size, as an indicator of prenatal undernutrition, has significant associations with two critical domains of healthy aging: physical and cognitive function. Despite the potential for bias, these results carry important public health implications. Individuals born with low birth weight should be acknowledged as a vulnerable population within the Public Health sector, which requires comprehensive follow-up and targeted interventions. Recognizing this group is essential for implementing effective support measures that address their unique needs. To mitigate the long-term risks associated with prenatal undernutrition, it is imperative to develop and sustain lifelong strategies. Additionally, ensuring access to quality nutrition during gestation and throughout an individual’s life is crucial for promoting optimal health outcomes. This can be achieved by fostering environments that support balanced diets and provide essential nutritional resources. Lifelong strategies aimed at mitigating the risks associated with prenatal undernutrition are crucial for promoting healthier aging trajectories.

## Figures and Tables

**Figure 1 nutrients-17-02583-f001:**
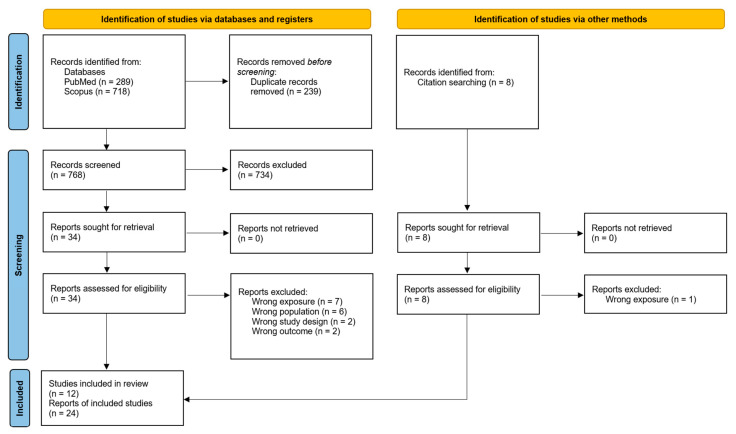
PRISMA flow chart for database searches and screening.

**Figure 2 nutrients-17-02583-f002:**
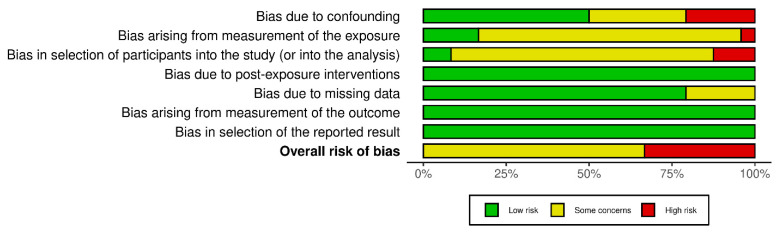
Summary of risk of bias per domain according to the RoB assessment.

**Figure 3 nutrients-17-02583-f003:**
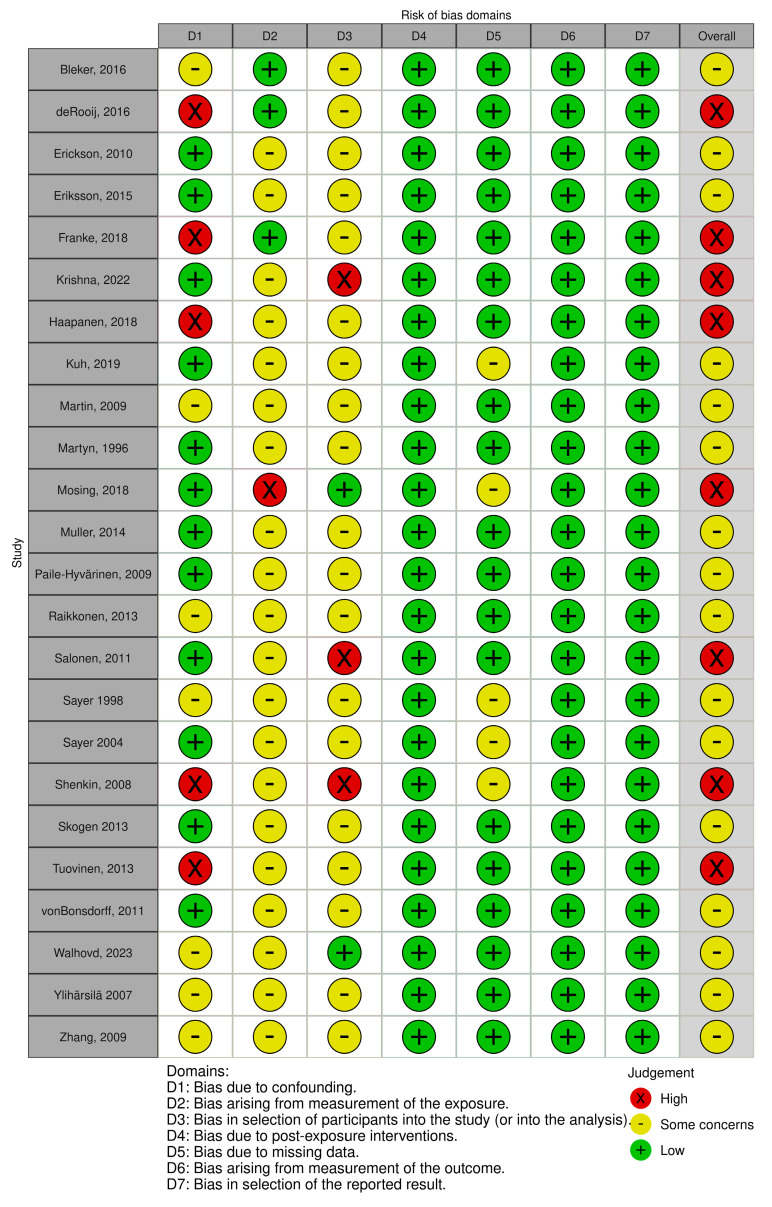
Risk of bias per domain for all included articles.

**Figure 4 nutrients-17-02583-f004:**
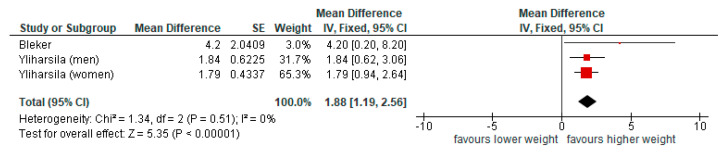
Pooled associations of birth weight and grip strength, unadjusted models (results presented on a scale from −10 to 10 kg GS).

**Figure 5 nutrients-17-02583-f005:**
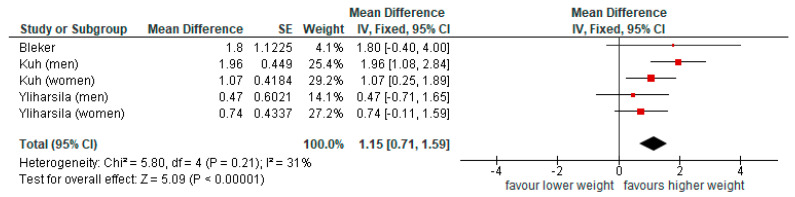
Pooled associations of birth weight and grip strength adjusted models (results presented on a scale from −4 to 4 kg GS).

**Figure 6 nutrients-17-02583-f006:**
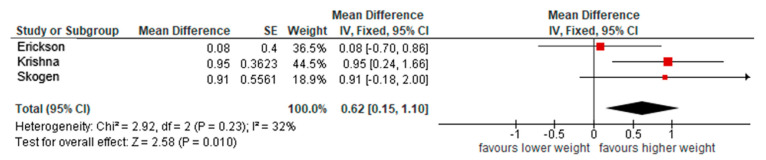
Pooled associations of birth weight and verbal fluency (results presented on a scale from −1 to 1 words/min).

**Table 1 nutrients-17-02583-t001:** Overview of included birth cohorts.

Cohort, Country	Reference(s)	No. of Participants	Population	Outcome Type
Helsinki Birth Cohort, Finland	Paile-Hyvärinen et al. [27] von Bonsdorff et al. [28] Salonen et al. [29] Raikkonen et al. [30] Tuovinen et al. [31] Eriksson et al. [32] Ylihärsilä et al. [22] Haapanen et al. [33]	13,345	Men and women born in Helsinki between 1934 and 1944 and were still living in Finland in 1971.	Cognitive Physical
Dutch Famine Birth Cohort, The Netherlands	Bleker et al. [20] deRooij et al. [34] Franke et al. [35]	2414	Men and women born in the Wilhelmina Gashuis in Amsterdam between 1 November 1943 and 28 February 1947.	Cognitive Physical
Hertfordshire Cohort, England	Martyn et al. [36] Sayer et al. [37] Sayer et al. [38] Martin et al. [39]	1576 1428 3207 1790	Men and women born in Herefordshire between; 1920 and 1943. 1920 and 1930. 1931 and 1939. 1931 and 1939.	Cognitive Physical
MRC NSHD ^a^ England	Kuh et al. [21]	5362	Sample of all births (men and women) in the first week of March in 1946 in mainland Britain.	Cognitive Physical
Edinburgh, Scotland	Shenkin et al. [40]	130	Community-dwelling volunteers born in one hospital in Edinburgh between 1921 and 1926.	Cognitive
Rancho Bernardo Study, United States	Erickson et al. [23]	2040	Men and women living in Rancho Bernardo, California in 1972 to 1974.	Cognitive
MYNAH Cohort ^b^, India	Krishna et al. [24]	3427	Men and women born in the Holdsworth Memorial Hospital in Mysore, South India between 1934 and 1966.	Cognitive
Swedish Twin Registry, Sweden	Mosing et al. [41]	35,357	All twins registered with the Swedish Twin Registry born between 1 January 1926 and 31 December 1960.	Cognitive
AGES ^c^ Cohort, Iceland	Muller et al. [42]	5764	Men and women born between 1907 and 1935 and living in Reykjavik in 1967.	Cognitive
HUSK ^d^ Cohort, Norway	Skogen et al. [25]	2156	Men and women of a previously established cohort living in Bergen/nearby Areas, born between 1925 and 1927.	Cognitive
Chinese Birth Cohort, China	Zhang et al. [43]	2062	Men and women born in Peking Union Medical College Hospital between 1921 and 1954.	Cognitive
UK Biobank, United Kingdom	Walhovd et al. [44]	500,000	Men and women living in the United Kingdom, aged between 40 to 69 years old in 2006 to 2010 and living close enough to an assessment center.	

^a^ Medical Research Council National Survey of Health and Development Cohort ^b^ MYsore studies of Natal effect on Ageing and Health Cohort ^c^ Age Gene/Environment Susceptibility—Reykjavik Study ^d^ The Hordaland Health Study.

**Table 2 nutrients-17-02583-t002:** Characteristics of included articles.

Author, Year	Participants (*n*)/Women (*n*)	Age (Mean, SD) Age Range	Exposure	Mean (SD)/Definition		Outcome/s	Confounders
**Physical function outcomes (*n* = 10)**							
von Bonsdorff, 2011 [28]	1999/1072	61.6 (2.9) 57 to 70	Low Birth Weight	<2.5 kg 2.5–2.9 kg vs. 3.0–3.4 kg		SF36 physical functioning questionnaire.	Sex, age, length of gestation, adult lean body mass, highest social class in childhood and adulthood, smoking status.
Eriksson, 2015 [32]	1078/603	71.3 (NA) 67 to 79	Birth Weight Birth Length Birth BMI	Men 3.5 (0.5) 50.7 (2.0) 13.5 (1.3)	Women 3.4 (0.4) 50.1 (1.8) 13.4 (1.2)	SFT Score, Chair Stands, Arm Curls, Chair Sit and reach, 6 min Walk, Back Scratch	Age, sex, occupation, education, adult lifestyle variables, metabolic measures, adult anthropometry.
Salonen, 2011 [29]	581/312	61.8 (2.8) NA	Birth Weight Birth Length Birth BMI	Men 3.5 (0.5) 50.8 (2.1) 13.5 (1.2)	Women 3.4 (0.4) 50.2 (1.8) 13.4 (1.2)	VO_2max_	Age, sex, adult lean body mass, adult social class, exercise and smoking habits.
Martin, 2009 [39]	629/280	67.9 (2.5) NA	Birth Weight	Men 3.5 (0.5)	Women 3.4 (0.5)	3-m walk, Chair Stands Balance	Age, height and weight adjusted for height.
Bleker, 2016 [20]	150/83	68 (NA) NA	Birth Weight	3.4 (0.5)		Grip Strength, SPPB, Frailty	Sex, BMI, SES, smoking, medication use, LTPA
Kuh, 2019 [21]	2098/1062	60.5 ^a^ (NA) 53 to 69	Birth Weight	Men 3.5 (0.5)	Women 3.3 (0.5)	Grip Strength	Age term, adult height.
Sayer, 1998 [37]	717/306	67.5 (2.3) NA	Birth Weight	Men 3.5 (0.5)	Women 3.4 (0.5)	Grip Strength	Age, sex, current social class, social class at birth and height
Sayer, 2004 [38]	1403/673	64.9 (2.6) NA	Birth Weight	Men 3.5 (0.5)	Women 3.4 (0.5)	Grip Strength	Age, physical activity, social class, smoking, and alcohol.
Ylihärsila, 2007 [22]	2003/1075	61.5 (2.9) 57 to 70	Birth Weight	Men 3.5 (0.5)	Women 3.4 (0.5)	Grip Strength	Age and adult BMI.
Haapanen, 2018 [33]	1078/603	71 (2.7) NA	Birth Weight Birth length Birth BMI	3.4 (0.5) 50.3 (1.9) 13.4 (1.2)		Pre-frailty Frailty	Age, sex, gestational age, childhood and adult SES, adult BMI, smoking, hypertension and diabetes.
**Cognitive function outcomes (*n* = 14)**							
de Rooij, 2016 [34]	118/66	67.5 (0.9) 65 to 69	Birth Weight	3.4 (0.5)		ICV, TBV	None reported.
Franke, 2018 [35]	52/0	67.5 (0.9) 65 to 69	Birth Weight	3.4 (0.5)		BrainAge Score	None reported.
Martyn, 1996 [36]	1576/Not reported	60.9 (2.1) 48 to 74	Birth Weight Birth Length Ponderal Index	3.4 (0.5) 51.1 (2.5) 25.4 kg/m^3^		Alice Heim intelligence test, Decline in Cognitive Function	Age, social class at birth and individual dataset.
Shenkin, 2009 [40]	128/90	78.4 (1.4) 75 to 81	Birth Weight Birth Length	3.3 (0.5) 50.6 (2.7)		National Adult Reading Test (NART), General cognitive factor (g), Raven’s Matrices Test, Murray House Test, Controlled Word Association, Logical Memory	Sex, gestational age, maternal Age, parity and social class
Erickson, 2010 [23]	292/292	71.1 (8.6) 55 to 89	Birth Weight	Not reported		Buschke Total Recall, Buschke LTM, Buschke STMa, Heaton Visual Copying, Heaton Visual LTM, Heaton Visual STM, MMSE, Serial 7’s, World backward, Trails B, Category fluency, Total blessed	Age and education
Krishna, 2022 [24]	721/313	62.3 (5.3) 55 to 80	Birth Weight Birth Length Ponderal Index	Men 2.8 (0.4) 48.2 (2.1) 25.4 (4.6)	Women 2.7 (0.4) 47.7 (2.9) 25.4 (5.1)	Global Cognition (SD) Verbal Fluency Score Immediate Recall Score Delayed Recall Score Composite Cognitive Score (SD)	Age, sex and sibship
Mosing, 2018 [41]	4000/2124	68.4 (2.6) 55 to 89	Birth Weigh Birth Length Low Birth Weight Small for Gestational Age	2.8 (0.5) ≤2.5 kg vs. >2.5 kg 2 SD below mean for a given GA and sex.		Cognitive Impairment	Age, sex, age of mother, parity, birth SES and education.
Muller, 2014 [42]	1254/715	75 (5) 69 to 82	Birth Weight Birth Length Ponderal Index	3.7 (0.5)		ICV, TBV, WMV, GMV CSFV, Ln WMLV Memory, Processing speed, Executive function	Sex, age, education, midlife weight, height, ICV.
Paile Hyvärinen, 2009 [27]	1243/658	63.9 (2.9) NA	Birth Weight	3.4 (0.5)		Divided Attention Task (DA) reaction time, Associate Learning Task (AL) hit rate	Age, sex, gestational age, CHD, DM, depressive symptoms and education.
Raikkonen, 2013 [30]	931/0	67.9 (2.5) 51 to 70	Birth Weight Birth Length Ponderal Index	3.5 (0.5) 50.7 (2.0) 26.6 (2.1)		Standardized Cognitive Ability (CA) Test Score, Decline in CA, Top third CA Score (vs. middle/bottom)	Length of gestation, father’s occupational status in childhood, parity, mother’s age and height at delivery, history of breastfeeding, age, educational level, stroke and CAD.
Tuovinen, 2013 [31]	876/377	69.3 (3.1) NA	Birth Weight	3.4 (0.5)		Complaints of cognitive failure, Dysexecutive functioning	Not reported.
Skogen, 2013 [25]	346/186	72.3 72 to 74	Birth Weight Birth Length Ponderal Index	3.5 (0.5) 50.3 (2.1) 27.3 (0.2)		Composite score, MMS, Digit symbol, KOLT, COWAT, TMA, Block design	Age and sex
Zhang, 2009 [43]	2062/1051	62.5 ^a^ (NA) 50 to 82	Ponderal Index	25.8 kg/m^2^		Lower cognition	Sex, gestational age, prenatal and early life factors
Walhovd, 2024 [44]	1759/1009	62 (7.1) 47 to 80.3	Birth Weight	3.4 (0.6)		Cortical volume	None reported.

Birth Weight is reported in kilograms (kg) and Birth Length in centimeters (cm). ^a^ Calculated weighted mean age.

**Table 3 nutrients-17-02583-t003:** Summary of findings for birth size and physical function.

Outcomes (Grouped)	Author [Ref]	Birth Size Exposure					
		Birth Weight		Birth Length		Birth BMI	
		*n* *		*n* *		*n* *	
Grip strength	Bleker [20], Kuh [21], Sayer [37], Sayer [38], Ylihärsila [22]	5	↑ (*n* = 4) ↔ (*n* = 1)	0		0	
Performance score	Bleker [20], Eriksson [32]	2	↑ (*n* = 1) ↔ (*n* = 1)	1	↑ (*n* = 1)	1	↑ (*n* = 1)
Walking	Martin [39], Eriksson [32]	2	↔ (*n* = 2)		↔ (*n* = 2)		↔ (*n* = 2)
Chair rise	Martin [39], Eriksson [32]	2	↔ (*n* = 1) ↔ (m ^a^)/↓ (w ^b^) (*n* = 1)	1	↔ (*n* = 1)	1	↔ (*n* = 1)
Balance	Martin [39]	1	↑ (m ^a^)/↔ (w ^b^) (*n* = 1)				
Arm curl	Eriksson [32]	1	↔ (*n* = 1)	1	↑ (*n* = 1)	1	↔ (*n* = 1)
Lower flexibility (sit and reach)	Eriksson [32]	1	↑ (*n* = 1)	1	↑ (*n* = 1)	1	↔ (*n* = 1)
Upper flexibility (back scratch)	Eriksson [32]	1	↔ (*n* = 1)	1	↔ (*n* = 1)	1	↑ (*n* = 1)
VO_2max_	Salonen [29]	1	↔ (*n* = 1)	1	↔ (*n* = 1)	1	↔ (*n* = 1)
Self-reported function	von Bonsdorff [28]	1	↑ (*n* = 1)				
Frailty	Haapanen [33]	1	↑ (*n* = 1)	1	↑ (*n* = 1)	1	↑ (*n* = 1)

↑ = positive association, ↔ = no significant association, ↓ = negative association, * *n* = number of measurements extracted per exposure ^a^ Men, ^b^ Women.

**Table 4 nutrients-17-02583-t004:** Summary of findings for birth size and cognitive function.

Outcomes (Grouped)	Author [Ref]	Birth Size Exposure									
		Birth Weight		Birth Length		Ponderal Index		SGA ^a^		LBW ^b^	
		*n* *		*n* *		*n* *		*n* *		*n* *	
Cognitive impairment	Tuovinen [31], Mosing [41], Skogen [25], Erickson [23]	6	↔ (*n* = 6)	2	↔ (*n* = 3)	2	↔ (*n* = 2)	1	↑ (*n* = 1)	1	↔ (*n* = 1)
Executive function	Skogen [25], Erickson [23], Muller [42]	2	↔ (*n* = 1)	1	↔ (*n* = 1)	2	↔ (*n* = 2)	0		0	
Word fluency	Krishna [24], Erickson [23], Shenkin [40]	4	↑ (*n* = 1) ↔ (*n* = 3)	3	↔ (*n* = 3)	2	↔ (*n* = 2)	0		0	
Cognition	Krishna [24], Shenkin [40], Skogen [25], Zhang [43]	5	↑ (*n* = 1) ↔ (*n* = 3)	4	↑ (*n* = 1) ↔ (*n* = 3)		↔ (*n* = 3)	0		0	
Verbal memory	Krishna [24], Erickson [23], Shenkin [40], Muller [42]	4	↑ (*n* = 2) ↔ (*n* = 2)	3	↑ (*n* = 1) ↔ (*n* = 2)	3	↔ (*n* = 3)	0		0	
Brain volume	Muller [42], de Rooij [34], Franke [35], Walhovd [44]	10	↑ (*n* = 4) ↔ (*n* = 6)	7	↑ (*n* = 1) ↔ (*n* = 6)	7	↑ (*n* = 5) ↔ (*n* = 2)	0		0	
Processing speed	Muller [42], Skogen [25], Paile-Hyvärinen [27]	3	↔ (*n* = 1) ↓ (*n* = 2)	1	↔ (*n* = 1)	2	↔ (*n* = 2)	0		0	
Cognitive decline	Raikkonen [30], Martyn [36]	2	↔ (*n* = 2)	2	↑ (*n* = 1) ↔ (*n* = 1)	2	↔ (*n* = 2)	0		0	
Simple attention	Skogen [25], Erickson [23]	4	↑ (*n* = 1) ↔ (*n* = 3)	1	↔ (*n* = 1)	1	↔ (*n* = 1)	0		0	
Intelligence	Martyn [36], Raikkonen [30], Shenkin [40]	4	↑ (*n* = 1) ↔ (*n* = 3)	4	↑ (*n* = 1) ↔ (*n* = 2)	2	↔ (*n* = 2)	0		0	
Premorbid ability	Shenkin [40]	1	↔ (*n* = 1)	0	↑ (*n* = 1)	0		0		0	
Visual memory	Erickson [23]	1	↔ (*n* = 1)	0		0		0		0	

↑ = positive association, ↔ = no significant association, ↓ = negative association, * *n* = number of measurements extracted per exposure ^a^ Small for Gestational Age ^b^ Low Birth Weight.

## Data Availability

Not applicable.

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
