# Peer review of "Associations of Birth Size with Physical and Cognitive Function in Men and Women 60 Years and Older—Systematic Review and Meta-Analysis"

_nutrients, 2025, doi:10.3390/nu17162583_

Round 1

Reviewer 1 Report

Comments and Suggestions for Authors

The text and abstract need to recognize that their exposure: “birth size as a measure of prenatal undernutrition” is limited since prenatal morbidities are highly influential to birth size and prenatal nutrition is strongly associated with the determinants of health (Carrillo-Alvarez PMID: 40334986).

Physical and cognitive function are also both highly influenced by the determinants of health (DOH) (https://www.who.int/health-topics/social-determinants-of-health), so this study’s analysis is highly confounded by the DOH. Thus the conclusion that “Small birth size, reflecting prenatal undernutrition, is associated with reduced physical and cognitive function in later life” is not likely valid.

Birth size is an intermediate variable. When found associated with outcomes, it is necessary to identify what aspect led to lower growth, was it prenatal growth, determinants of health, nutrition, morbidities, stress, etc.

Assessing the effect of adjusting for sex, age and height could be of interest and it is valuable when possible to report both unadjusted and adjusted to be able to see the effect of the adjustment. Of importance, it is not valid to adjust for body mass index in analysis of outcomes that are associated with body size as it causes an over-adjustment. For this systematic review to be valid, the results need to be separated by those adjusted for body weight or BMI (and considered over-adjusted) versus those not adjusted for body weight or BMI.

References regarding over-adjustment by body weight:

  1. Kramer MS, Zhang X, Dahhou M, Yang S, Martin RM, Oken E, et al. Does fetal growth restriction cause later obesity? Pitfalls in analyzing causal mediators as confounders. Am J Epidemiol. 2017 Apr;185(7):585–90. 28338874
  2. Huxley R, Neil A, Collins R. Unravelling the fetal origins hypothesis: is there really an inverse association between birthweight and subsequent blood pressure? Lancet 2002 Aug;360(9334):659–65. 12241871
  3. Ananth C V., Schisterman EF. Confounding, causality, and confusion: the role of intermediate variables in interpreting observational studies in obstetrics. Am J Obstet Gynecol [Internet]. 2017;217(2):167–75.
  4. Schisterman EF, Cole SR, Platt RW. Overadjustment bias and unnecessary adjustment in epidemiologic studies. Epidemiology [Internet]. 2009;20(4):488–95.
  5. Paneth N, Ahmed F, Stein AD. Early nutritional origins of hypertension: a hypothesis still lacking support. J Hypertens Suppl. 1996 Dec;14(5):S121–9.
  6. Elmrayed S, Metcalfe A, Brenner D. Wollny K. & Fenton TR. Are small-for-gestational-age preterm infants at increased risk of overweight? Statistical pitfalls in overadjusting for body size measures. J Perinatol (2021). PMID: 33850286 https://rdcu.be/ciB7k
  7. Williams TC, Bach CC, Matthiesen NB, Henriksen TB, Gagliardi L. Directed acyclic graphs: a tool for causal studies in paediatrics. Pediatr Res. 2018 Oct;84(4):487-493. PMID: 29967527; PMCID: PMC6215481.

It would be appropriate to consider the “studies [that] made further adjustments for education and/or socio-economic status” as the superior adjustments.

Author Response

Manuscript ID: nutrients-3767544

Response to Reviewer 1 Comments

  1. Summary

Thank you very much for taking the time to review this manuscript. We have carefully revised all comments and provided answers below. Please find the detailed responses below, along with the corresponding revisions/corrections highlighted in track changes in the resubmitted files.

We hope that this review will be the first of many papers published to highlight the importance of recognizing small-born individuals as a vulnerable group and to advocate for policy support for maternal health.

  1. Questions for General Evaluation Reviewer’s Evaluation Response and Revisions

Does the introduction provide sufficient background and include all relevant references?          Yes/Can be improved/Must be improved/Not applicable             

Response: Authors inserted clarification line 46-47, page 2 (in red), to answer reviewers' comment 1. Changes were made to the text (in yellow) to make it more transparent and readable.

Are all the cited references relevant to the research?            Yes/Can be improved/Must be improved/Not applicable 

Response: Since no comments are provided regarding improper or better references for the paper, we are unable to respond to this. We have a maximum number of references, which means that we are unable to include those references that Reviewer 1 pointed out. According to reviewer 2, the cited references are relevant.

Is the research design appropriate?            Yes/Can be improved/Must be improved/Not applicable        

Response: The authors rigorously followed the PRISMA statement and the ROBINS-E tool to evaluate the risk of bias. Since no comments are provided regarding improper or better references for the paper, we are unable to respond to this. According to reviewer 2, the research design is appropriate.

Are the methods adequately described?     Yes/Can be improved/Must be improved/Not applicable        

Response: Since no comments are provided regarding improper or better references for the paper, we are unable to respond to this. According to reviewer 2, the methods are appropriate.

Are the conclusions supported by the results? Yes/Can be improved/Must be improved/Not applicable

Response: By reviewing and addressing the reviewers' comments, we aim to strengthen the support for the conclusion based on the results.

Are all figures and tables clear and well-presented? Yes/Can be improved/Must be improved/Not applicable

Response: The caption for Figure 2, page 15, in line 195 was revised to clarify the figure's content, allowing it to stand alone.

  1. Point-by-point response to Comments and Suggestions for Authors

Comment 1: The text and abstract need to recognize that their exposure: “birth size as a measure of prenatal undernutrition” is limited since prenatal morbidities are highly influential to birth size and prenatal nutrition is strongly associated with the determinants of health (Carrillo-Alvarez PMID: 40334986).

Response 1: Thank you for bringing this to our attention. We agree with this comment and, therefore, have inserted clarification lines 46-47 on page 2 (in red) to clarify this for the reader. As the research question in this systematic review and meta-analysis is whether smaller birth size, as indicated by prenatal undernutrition for any reason, is associated with reduced physical and cognitive function in individuals aged 60 years and older, we aim to focus on the question, rather than the reasons.

Comment 2: Physical and cognitive function are also both highly influenced by the determinants of health (DOH) (https://www.who.int/health-topics/social-determinants-of-health), so this study’s analysis is highly confounded by the DOH. Thus the conclusion that “Small birth size, reflecting prenatal undernutrition, is associated with reduced physical and cognitive function in later life” is not likely valid.

Response 2: We agree that the determinants of health (DOH) play a significant role in influencing both physical and cognitive function. However, social factors and other variables, such as prenatal undernutrition, further complicate the understanding of DOH. As mentioned in the chapter “Strength and Limitations,” we made efforts to minimize the impact of gestational age at birth, and it is noteworthy that nearly all articles included in our analysis originated from high-income countries, apart from two (see lines 167-168 on page 4 and Table 1 on page 5).

Comment 3: Birth size is an intermediate variable. When found associated with outcomes, it is necessary to identify what aspect led to lower growth, was it prenatal growth, determinants of health, nutrition, morbidities, stress, etc.

Response 3: We have modified the chapter “Strengths and Limitations” to emphasize this point with the sentence “Using birth size as a measure of prenatal undernutrition is a limitation due to its indirect nature and influence by multiple factors”, lines 362-363 on page 15.

As mentioned in the limitations section (page 15) of our discussion chapter, we acknowledge potential limitations to using birth size as a measure of prenatal undernutrition.

A possible limitation is that the exposure of interest, prenatal undernutrition, is difficult to measure; therefore, we used birth size. However, prenatal undernutrition can have varying effects on birth size, depending on factors such as the timing and duration of undernutrition. Additionally, several factors beyond prenatal undernutrition influence birth size. We attempted to minimize the effect of gestational age at birth by excluding studies on preterm-born participants and considering gestational age for other cohorts in the risk of bias assessment. While we acknowledge that the exposure of interest is not perfect, in our opinion, it is the best possible choice.”

Furthermore, the methodological design of the cohort studies included in the current systematic review aims to ensure that the included participants represent the entire population, including those with varying birth sizes. Further insights into studies included in the Meta-Analyses are provided below.

Studies included in the Meta-Analyses for grip strength.

  • Bleker et al. (2016) report that one-third of participants were exposed to prenatal undernutrition (Dutch famine), and that those participants generally had lower birth weight than those not exposed (20).
  • Kuh et al. (2019) don’t report the number of participants in specific birth weight groups or below a particular birth weight (21). However, previous publications from the same birth cohort (the 1946 British Birth Cohort) have reported that around 20% of participants had a birth weight of 3.0 kg or less (14).
  • Ylihärsilä et al. (2007) report around 20% of participants having a birth weight of 3.0 kg or less (Helsinki Birth Cohort) (22).

Studies included in the Meta-Analyses for word fluency.

  • Erickson et al. (2010) report birth weight in tertiles, with the lowest tertile including participants with a birth weight of 3.1 kg or lower (United States) (32).
  • Krishna et al. (2022) report that over a quarter of the participants had birth weights below 2.5 kg, in a population of generally lower birth weight than other included cohorts (from South India) (24).
  • Skogen et al. (2013) don’t report the number of participants in specific birth weight groups or below a particular birth weight (Norway) (25). Results show an average birth weight of 3.47 kg with a standard deviation of 0.53 kg. The authors acknowledge a lack of statistical power to analyze results based on specific cut-off points for birth weight (i.e., <2.5 kg), given a sample size of n = 346.

Therefore, we stand by the current results of our manuscript, which examine the association between birth size and physical and cognitive function. Considering birth size, a measure of prenatal nutrition, where those born generally smaller than their peers may have been undernourished.

Comment 4: Assessing the effect of adjusting for sex, age and height could be of interest and it is valuable when possible to report both unadjusted and adjusted to be able to see the effect of the adjustment. Of importance, it is not valid to adjust for body mass index in analysis of outcomes that are associated with body size as it causes an over-adjustment. For this systematic review to be valid, the results need to be separated by those adjusted for body weight or BMI (and considered over-adjusted) versus those not adjusted for body weight or BMI.

Response 4:

Thank you for your comment. As shown in several studies, higher BMI correlates with poorer performance on tests such as the six‐minute walk, chair stand, and gait speed. For example, Schoffman et al. (2013) and Hergenroeder et al. (2011) report that failing to adjust for elevated BMI may misrepresent the relationship between body composition and function; the latter noted a roughly 50% attenuation in the association between self‐reported and performance-based measures when BMI was accounted for. Jankowski et al. (2008) found that BMI is a strong predictor of function (explaining up to 50% of variability), while Liao et al. (2017) document that obesity significantly worsens knee flexion and self-reported function among total knee replacement patients.

These studies, among others, support the use of BMI adjustment in clinical assessments to ensure the validity of physical function outcomes.

  • Danielle E. Schoffman, S. Wilcox, M. Baruth. Arthritis. 2013 https://doi.org/10.1155/2013/190868 Associations between BMI and all functional measures remained significant in the adjusted models (s ≤ 0.001).
  • Jankowski, W. Gozansky, R. E. Pelt, M. Schenkman, P. Wolfe, R. Schwartz, W. Kohrt. Obesity. 2008. https://doi.org/10.1038/oby.2007.84 Adiposity was a stronger predictor of measured and self‐reported physical function than was muscularity in older adults living independently. BMI, adjusted for sex, is a reasonable substitute for adiposity in the prediction of physical function.
  • Şavkın R, Bayrak G, Büker N. The effects of the body mass index on the physical function and the quality of life in the elderly. Balt J Health Phys Act. 2020; Suppl(1):55-62. doi: 10.29359/BJHPA.2020. BMI negatively affects the physical function, physical health, and the psychological domain of the quality of life in the elderly.
  • Woo J, Leung J, Kwok T. BMI, body composition, and physical functioning in older adults. Obesity (Silver Spring). 2007 Jul;15(7):1886-94. doi: 10.1038/oby.2007.223. PMID: 17636108. Those with BMI> 30 kg/m2 had the worst walking performance, and the groups with a BMI in the normal and overweight range had optimal performance. Fat mass, but not appendicular muscle mass, was associated with walking speed after adjusting for BMI. Fat mass seems to be a more critical factor than appendicular muscle mass in determining walking speed in community-living older adults, even after adjusting for BMI.
  • Nascimento, M.d.M.; Gouveia, É.R.; Gouveia, B.R.; Marques, A.; Campos, P.; García-Mayor, J.; Przednowek, K.; Ihle, A. The Mediating Role of Physical Activity and Physical Function in the Association between Body Mass Index and Health-Related Quality of Life: A Population-Based Study with Older Adults. Int. J. Environ. Res. Public Health 2022, 19, 13718. https://doi.org/10.3390/ijerph192113718 Correlation analyses showed that anthropometric variables (BMI, WC, WHR) were positively related to each other, and negatively and small with PA, PF, and HRQoL. On the other hand, WHR indicated a negative and medium correlation with HRQoL.

Information on confounders that were adjusted for in the included articles is provided in Table 2 in the Manuscript (line 192, page 15). Out of the ten included papers that investigated physical functioning outcomes, only one did not adjust for adult anthropometric measures, such as BMI or height (Sayer 2004). Thus, further separation of results could be considered unwieldy. Furthermore, the meta-analysis for grip strength reports both unadjusted and adjusted values, with all authors adjusting for anthropometric measures (Figures 4 and 5).

We assume that physical skills in later life may be closely related to body composition. Therefore, the claim that it is not possible to adjust for body size in statistical analyses when examining skills without overcorrecting must be questioned. It can be argued that results may be insignificant for outcomes such as skills if they are not adjusted for some measure of body size (as can be seen in the literature, that the vast majority of articles that include these measurements at all include such adjustments).

Regarding the included articles that investigated cognitive functioning, only one out of 14 (Muller et al, 2014) adjusted for adult anthropometric measures.

  1. Response to Comments on the Quality of English Language

Point 1: The English could be improved to more clearly express the research

Response 1: Language improvements were made throughout the paper, including the abstract and main text, to enhance overall clarity and readability. Changes are highlighted in yellow.

  1. Additional clarifications

None

Reviewer 2 Report

Comments and Suggestions for Authors

The manuscript addresses an important and timely topic with clear relevance to public health. The systematic review is well designed, follows the PRISMA guidelines, and the registration of the protocol on PROSPERO enhances transparency. The literature search is comprehensive, and the use of the ROBINS-E tool for bias assessment is appropriate. The methods, inclusion/exclusion, and assessment of bias risks are well described. The systematic review is well designed, follows the PRISMA guidelines, and the registration of the protocol on PROSPERO enhances transparency. The bibliographic search is comprehensive, and the use of the ROBINS-E tool for bias assessment is appropriate. The methods, inclusion/exclusion criteria and synthesis strategies are described in detail.

Major points:

  • Strengthen the discussion on the possible role of postnatal environmental and socio-economic factors as potential residual confounders.

  • Clarify the limitations of using birth weight as the sole indicator of prenatal nutrition, specifying the possible influence of other maternal or neonatal factors.

  • Improve clarity on the management of heterogeneity of outcome measures across included studies.

  • Specify, if possible, the criteria used for assessing risk of bias in individual ROBINS-E domains.

  • Add a brief reflection on the practical implications of the results, especially in terms of prevention and health policies.

Minor points:

  • Briefly expand on the practical implications of the findings, particularly for prevention and public health strategies.

  • Minor language improvements could be considered to enhance overall clarity.

Author Response

Manuscript ID: nutrients-3767544

Response to Reviewer 2 Comments

  1. Summary

Thank you very much for taking the time to review this manuscript and for your positive response. We have carefully revised all comments and provided answers below. Please find the detailed responses below, along with the corresponding revisions/corrections highlighted in track changes in the resubmitted files.

  1. Questions for General Evaluation Reviewer’s Evaluation Response and Revisions

Does the introduction provide sufficient background and include all relevant references?          Yes/Can be improved/Must be improved/Not applicable             

Response: Authors inserted clarification line 46-47, page 2 (in red), to answer reviewers' comment 1. Changes were made to the text (in yellow) to make it more transparent and readable.

Are all the cited references relevant to the research?            Yes/Can be improved/Must be improved/Not applicable 

Is the research design appropriate?            Yes/Can be improved/Must be improved/Not applicable        

Are the methods adequately described?     Yes/Can be improved/Must be improved/Not applicable        

Are the conclusions supported by the results? Yes/Can be improved/Must be improved/Not applicable

Response: By reviewing and addressing comments 2, 3, and 4, we hope to support the conclusion with the results.

Are all figures and tables clear and well-presented? Yes/Can be improved/Must be improved/Not applicable

Response: The caption for Figure 2, page 15, in line 195 was revised to clarify the figure's content, allowing it to stand alone. Other Figures and Tables seem to be according to standards.

  1. Point-by-point response to Comments and Suggestions for Authors

The manuscript addresses an important and timely topic with clear relevance to public health. The systematic review is well designed, follows the PRISMA guidelines, and the registration of the protocol on PROSPERO enhances transparency. The literature search is comprehensive, and the use of the ROBINS-E tool for bias assessment is appropriate. The methods, inclusion/exclusion, and assessment of bias risks are well described. The systematic review is well designed, follows the PRISMA guidelines, and the registration of the protocol on PROSPERO enhances transparency. The bibliographic search is comprehensive, and the use of the ROBINS-E tool for bias assessment is appropriate. The methods, inclusion/exclusion criteria and synthesis strategies are described in detail.

Comment 1: Strengthen the discussion on the possible role of postnatal environmental and socio-economic factors as potential residual confounders.

Response 1: Thank you for bringing this to our attention. We agree with this comment and, therefore, have inserted clarification lines 46-47 on page 2 (in red) to clarify this for the reader in the Background section. As the research question in this systematic review and meta-analysis is whether smaller birth size, as indicated by prenatal undernutrition for any reason, is associated with reduced physical and cognitive function in individuals aged 60 years and older, we aim to focus on the question, rather than the reasons.

Comment 2: Clarify the limitations of using birth weight as the sole indicator of prenatal nutrition, specifying the possible influence of other maternal or neonatal factors.

Response 2: To clarify the limitation of using birth weight as the sole indicator of prenatal nutrition, we added a sentence in the “Strengths and Limitations” section on page 14, lines 367-368, in red.

Comment 3: Improve clarity on the management of heterogeneity of outcome measures across included studies.

Response 3: For improved clarity regarding heterogeneity of outcome measures, we rewrote the section in the “Strengths and Limitations” section, page 17, lines 377-381, in yellow.

Comment 4: Specify, if possible, the criteria used for assessing risk of bias in individual ROBINS-E domains.

Response 4: Thank you for your comment. We have rewritten this section for greater clarity. See page 9, lines 198-205, in red.

Comment 5: Add a brief reflection on the practical implications of the results, especially in terms of prevention and health policies.

Response 5: We have rewritten in the “Discussion” section a practical implication in public health monitoring, page 15, lines 347-352, in red.

Comment 6: Briefly expand on the practical implications of the findings, particularly for prevention and public health strategies.

Response 6: Practical implications were rewritten and inserted in the “Conclusion” chapter, page 15, lines 392-400 in red.

  1. Response to Comments on the Quality of English Language

Point 1: Minor language improvements could be considered to enhance overall clarity.

Response 1: Language improvements were made throughout the paper, including the abstract and main text, to enhance overall clarity and readability. Changes are highlighted in yellow.

  1. Additional clarifications

None

Round 2

Reviewer 2 Report

Comments and Suggestions for Authors

tha authors have addressed all my concerns